# ROBIN: Robust and Invisible Watermarks for Diffusion Models with Adversarial Optimization

**Huayang Huang**[1], **Yu Wu**[1,*], **Qian Wang**[2]
[1]School of Computer Science, Wuhan University
[2]School of Cyber Science and Engineering, Wuhan University
{hyhuang,wuyucs,qianwang}@whu.edu.cn

## Abstract

Watermarking generative content serves as a vital tool for authentication, ownership protection, and mitigation of potential misuse. Existing watermarking methods face the challenge of balancing robustness and concealment. They empirically inject a watermark that is both invisible and robust and *passively* achieve concealment by limiting the strength of the watermark, thus reducing the robustness. In this paper, we propose to explicitly introduce a watermark hiding process to *actively* achieve concealment, thus allowing the embedding of stronger watermarks. To be specific, we implant a robust watermark in an intermediate diffusion state and then guide the model to hide the watermark in the final generated image. We employ an adversarial optimization algorithm to produce the optimal hiding prompt guiding signal for each watermark. The prompt embedding is optimized to minimize artifacts in the generated image, while the watermark is optimized to achieve maximum strength. The watermark can be verified by reversing the generation process. Experiments on various diffusion models demonstrate the watermark remains verifiable even under significant image tampering and shows superior invisibility compared to other state-of-the-art robust watermarking methods.

## 1 Introduction

Diffusion models (DMs) are revolutionizing content creation and generating stunningly realistic imagery across diverse domains [17, 33, 60]. The advent of text-to-image diffusion models [31, 30, 58], coupled with personalized generation techniques [53, 7, 32, 15, 41, 59], enables the creation of highly specific content by virtually anyone. However, it has raised concerns about authenticity and ownership, including the risk of plagiarism [34, 22] and the potential misuse of images of public figures [39, 5]. Consequently, governments and businesses are increasingly advocating for robust mechanisms to verify the origins of generative content [19, 45].

Watermarking offers a proactive approach to authenticate the source of generated content. This technique embeds imperceptible secret messages within the generated content. These messages serve as unique identifiers, confirming the image's origin while remaining invisible to the human eye. They also need to be robust enough to withstand potential distortions encountered during online sharing.

Existing watermarking techniques face a significant challenge in striking a balance between concealment and robustness. Traditional post-processing methods [46, 9] employ an empirical approach to identify an invisible and robust watermark and embed it within the generated image. They *passively* achieve concealment by limiting the watermark strength, consequently compromising robustness. Conversely, stronger watermarks, while enhancing robustness, can introduce visible artifacts into the generated image. Recent advancements in in-processing watermarking for diffusion models expect

---

*Corresponding author

38th Conference on Neural Information Processing Systems (NeurIPS 2024).

the generative model to learn this balance and directly produce watermarked content. However, these methods often require expensive model retraining [55, 48, 13] or can lead to unintended semantic alterations within the generated images [44].

Our ROBIN scheme introduces an explicit watermark hiding process to *actively* achieve concealment. This approach reduces the invisibility limitation of the watermark itself and thus enables the embedding of more robust watermarks. Specifically, we implant a robust watermark within an intermediate diffusion state, and then directionally guide the model to gradually conceal the implanted watermark, thus achieving invisibility in the final generated image. In this way, robust watermarks can be secretly implanted in the generated content without model retraining.

We focus on the text-to-image diffusion models, which support an additional prompt signal to guide the generation process. We employ an adversarial optimization algorithm to design an optimal prompt guidance signal specifically tailored for each watermark. **The prompt embedding is optimized to minimize artifacts in the generated image, and the watermark is optimized to achieve maximum strength.** The optimized watermark and prompt signal are universally applicable to all images. During the generation process, the watermark is implanted within an intermediate state following the semantic formation stage. Subsequently, the optimized prompt guidance signal is introduced throughout the remaining diffusion steps. After image generation, following previous works [44, 49], we reverse the diffusion process to the watermark embedding point to verify the existence of the watermark. This innovative approach offers a promising way to overcome the trade-off between watermark strength and stealth by explicitly introducing an additional watermark hiding process.

In summary, our key contributions are as follows:

- We propose a novel watermarking method for diffusion models that embed a robust watermark and subsequently employ an active hiding process to achieve imperceptibility.

- We develop an adversarial optimization algorithm to generate a prompt signal for watermark hiding and a strong watermark that can be hidden and strategically select the watermarking point within the diffusion trajectory.

- Evaluations on both latent and image diffusion models demonstrate that our scheme exhibits superior robustness against various image manipulations while preserving semantic content.

## 2 Related work

**Diffusion generation and inversion.** Diffusion models [17, 12, 36, 37] operate by iteratively transforming pure noise $x_T \sim \mathcal{N}(0, \mathbf{I})$ into increasingly realistic images $x_0 \sim q(x)$ through $T$ steps of denoising. The learning process involves a stochastic Markov chain in two directions. The forward process diffuses the sample $x_0$ by adding random noise:

$$q(x_t|x_{t-1}) = \mathcal{N}(\sqrt{1 - \beta_t}x_{t-1}, \beta_t \mathbf{I}), \tag{1}$$

where $\{\beta_t\}_{t=1}^T$ is the scheduled variance. $x_t$ can also be generated from $x_0$ as:

$$x_t = \sqrt{\bar{\alpha}_t}x_0 + \sqrt{1 - \bar{\alpha}_t}\epsilon, \tag{2}$$

where $\bar{\alpha}_t = \prod_{t=1}^T (1 - \beta_t)$ and $\epsilon \sim \mathcal{N}(0, \mathbf{I})$. Then a network $\epsilon_\theta$ is learned to predict the noise in each step, following the objective:

$$\min_\theta E_{x_0, t \sim \texttt{Uniform}(1,T), \epsilon \sim \mathcal{N}(0,\mathbf{I})} \|\epsilon - \epsilon_\theta(x_t, t, \psi(p))\|_2^2, \tag{3}$$

where $x_t$ is the noise latent at timesteps $t$ and $\psi(p)$ is the embedding of the text prompt $p$.

DDIM (Denoising Diffusion Implicit Model) [35] introduces the ODE solver for deterministic sampling by constructing the original one as a non-Markov process. It computes the $x_{t-1}$ from $x_t$ by predicting the estimation of $x_0$ and the direction pointing to $x_t$:

$$x'_0 = \frac{x_t - \sqrt{1 - \bar{\alpha}_t}\epsilon_\theta(x_t, t, \psi(p))}{\sqrt{\bar{\alpha}_t}}, \tag{4}$$

$$x_{t-1} = \sqrt{\bar{\alpha}_{t-1}}x_0' + \sqrt{1-\bar{\alpha}_{t-1}}\epsilon_\theta(x_t, t, \psi(p)). \tag{5}$$

The deterministic generation properties of DDIM allow it to reconstruct the noise latent $\hat{x}_t$ from the final image $x_0$ as :

$$\hat{x}_t = \sqrt{\bar{\alpha}_t}x_0 + \sqrt{1-\bar{\alpha}_t}\epsilon_\theta(x_{t-1}, t-1). \tag{6}$$

This unique characteristic allows us to selectively mark and recover an inner noise representation within the diffusion process, which serves as a powerful tool for our watermarking approach.

**Watermarking generative models.** The content watermark of generative models can be introduced either after the generation (post-processing) or during the sampling process (in-processing). Post-processing methods can adopt traditional digital image watermarking technology. Popular methods include frequency domain watermarking, which modifies the image representation in domains like Discrete Wavelet Transform (DWT) [47] or Discrete Cosine Transform (DCT) [8]. DwtDct watermarking [3] is applied in open sourced model Stable Diffusion. Frequency domain watermarks can be designed to be robust against common image manipulations like cropping, scaling, and even compression [38]. HiDDeN [57] pioneered the end-to-end approach, utilizing an encoder-decoder architecture to directly generate watermarked images. RivaGAN [52] leverages adversarial training to incorporate perturbations and image processing during model training for increased robustness.

In-processing methods make the watermark become part of the generated image by interfering with the generation process. Early approaches explored adding watermarks to training data [50, 55, 10, 13, 48], essentially building a watermark encoder into the model. Stable Signature [14] simplified this process by fine-tuning only the external decoder of latent diffusion models. However, these methods all treated watermarking as a separate goal from the generation task, limiting their flexibility. The recent Tree-Ring watermarking [44] shares similarities with our approach, modifying the initial noise to encode information semantically within the image. However, the semantic modifications induced by Tree-Ring watermarks are random and may compromise the faithfulness of the original model. Therefore, we aim to preserve the original semantics exactly to guarantee a similar level of text alignment compared to the original generation. Our work shows that embedding the watermark within the intermediate diffusion state and guiding the model to hide it can achieve the secret embedding of strong watermarks without model retraining.

## 3 Methodology

### 3.1 Overview of ROBIN

**Task definition.** Diffusion model watermarking aims to embed an invisible and verifiable watermark $w_i$ within the generated image $x_0$, using a watermark implantation function $I$. During Internet transmission, the generated content may be subjected to various image transformation operations $\mathcal{T}$. The model owner aims to leverage a watermark extraction algorithm $E$ to verify the presence of $w_i$ within the distorted sample $\mathcal{T}(x_0)$, thereby establishing image ownership.

**Pipeline of ROBIN.** *Watermark generation.* We first generate a hiding prompt guidance signal $w_p$ for each watermark $w_i$ using the adversarial optimization algorithm, which is detailed in Section 3.2.

*Watermark implantation.* ROBIN implants $w_i$ into an intermediate generation state $x_t$ after the semantics have been formed as

$$x_t^* = I(x_t, w_i, \mathbb{M}), \tag{7}$$

where $I$ injects $w_i$ into the frequency domain of $x_t$ and $\mathbb{M}$ is the coverage area of the watermark. During the remaining DDIM generation, ROBIN incorporates the optimized prompt guidance signal $w_p$ to direct the model towards hiding the watermark $w_i$ to maintain the similarity between the generated image $x_0^*$ and its unwatermarked counterpart $x_0$. Let $t_{\text{injection}}$ be the watermark injection point, the generation of the watermarked image is as follows:

$$p_\theta^{(t)}(x_{t-1}|x_t) = \begin{cases} \sqrt{\bar{\alpha}_{t-1}}x_0' + \sqrt{1-\bar{\alpha}_{t-1}}\epsilon_\theta(x_t, t, \psi(p)) & \text{if } T \geq t > t_{\text{injection}} \\ \sqrt{\bar{\alpha}_{t-1}}x_0'^* + \sqrt{1-\bar{\alpha}_{t-1}}\epsilon_\theta(x_t^*, t, \psi(p), w_p) & \text{if } t_{\text{injection}} \geq t \end{cases} \tag{8}$$

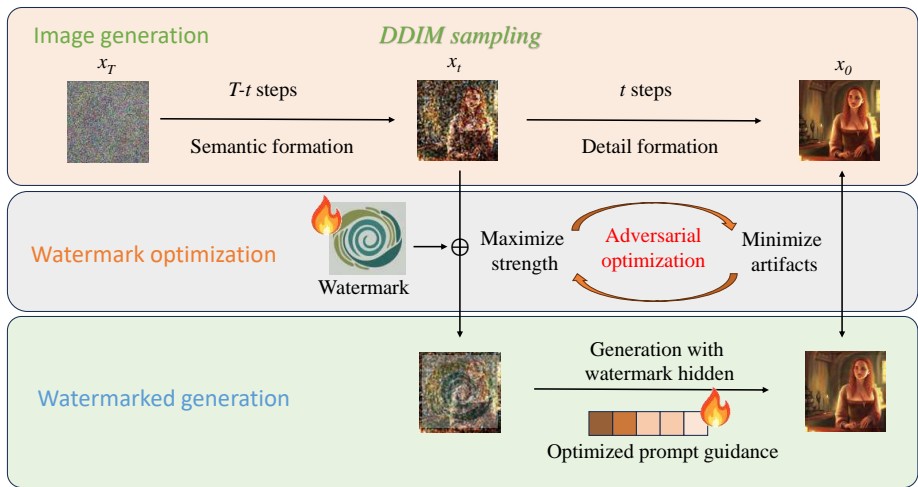

Figure 1: The watermark optimization and implantation of ROBIN. A robust watermark is added at an intermediate state of generation, and an additional prompt guiding signal is optimized to direct the model towards hiding the embedded watermark in the generated image. The image watermark and guiding signal are optimized adversarially to improve robustness and invisibility.

After embedding the watermark, the model is guided by both the original input text prompt $p$ and the optimized prompt embedding $w_p$ to achieve reliable generation with the watermark hidden. The predicted noise then becomes

$$\epsilon_\theta(x_t^*, t, \psi(p), w_p) = \eta_1 \cdot \epsilon_\theta(x_t^*, t, \psi(p)) + \eta_2 \cdot \epsilon_\theta(x_t^*, t, w_p) \tag{9}$$
$$+ (1 - \eta_1 - \eta_2) \cdot \epsilon_\theta(x_t^*, t, \psi(\emptyset)), \tag{10}$$

where $\eta_1, \eta_2$ are the guidance scale parameters to weight the guidance of the original text prompt and the optimized prompt signal.

*Watermark verification.* To verify the watermark, we reverse the transformed watermarked image $\mathcal{T}(x_0^*)$ to step $t_{\text{injection}}$ and retrieve the intermediate state $\hat{x}_t^*$. The watermark information $w' = E(\hat{x}_t^*, \mathbb{M})$ is extracted from the frequency space of $\hat{x}_t^*$. L1 distance $D$ is used to measure the similarity between $w$ and $w'$. When the distance falls below a threshold as

$$D(w, w') = \frac{1}{|\mathbb{M}|} \sum_{i \in \mathbb{M}} |w_i - w_i'| \leq \tau, \tag{11}$$

the presence of the watermark within the image is confirmed. Figure 1 presents the watermark generation and implantation process of ROBIN.

## 3.2 Adversarial optimization algorithm

We employ an adversarial optimization algorithm to generate the watermark and the corresponding hiding prompt guidance signal. The prompt signal is optimized in the embedding space and guides the model to conceal the embedded image watermark, while the watermark tries to be as strong as possible while allowing for its targeted hiding by the prompt signal.

The objective of the prompt guiding signal is to minimize the impact of the watermark on the final generated image. We define the image retaining loss $l_{ret}$, which penalizes excessive deviations from the original images:

$$\ell_{ret} = \text{MSE}(x_0'^* - x_0), \tag{12}$$

$$x_0'^* = \frac{x_t^* - \sqrt{1 - \bar{\alpha}_t}\epsilon_\theta(x_t^*, t, \psi(p), w_p)}{\sqrt{\bar{\alpha}_t}}. \tag{13}$$

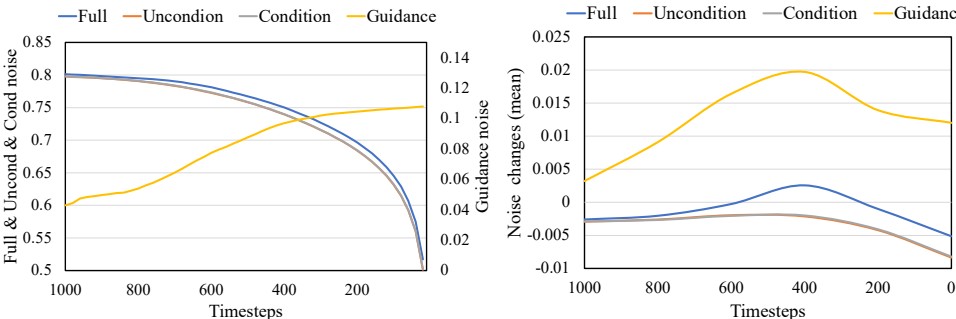

(a) Predicted noise during normal generation.  (b) Predicted noise changes under perturbations.

Figure 2: The impact of introducing frequency domain disturbances at different diffusion steps on the predicted noise. Timestep 1000 signifies the Gaussian noise state and step 0 represents the final generated image. The Uncondition curve (orange) and the Condition curve (gray) nearly overlap in both figures. Guidance is the amplified difference of Uncondition and Condition. Full is the addition of Uncondition and Guidance.

$x_0'^*$ is the final image predicted from the watermarked noisy latent $x_t^*$ through Equation (4) with an additional guidance $w_p$. MSE denotes the mean squared error.

Furthermore, as the loss incurred during DDIM inversion increases proportionally with the guidance strength [25], we introduce a constraint term $l_{cons}$ to prevent excessive prompt guidance:

$$\ell_{cons} = \text{MSE}(\epsilon_\theta(x_t^*, t, w_p) - \epsilon_\theta(x_t^*, t, \psi(\emptyset))). \tag{14}$$

To achieve robustness, we embed the watermark in the frequency domain of the image [44]. Frequency domain signals are more resistant to spatial operations compared to spatial domain signals [43]. Similar to [44], we set the watermark as multiple concentric rings, but we further optimize its value to the maximum within the aforementioned constraints for greater strength and better robustness. The optimization losses of $w_p$ and $w_i$ become

$$\mathcal{L}_{w_p} = \alpha\ell_{ret} + \beta\ell_{cons}, \tag{15}$$

$$\mathcal{L}_{w_i} = \alpha\ell_{ret} + \beta\ell_{cons} - \lambda\|w_i\|. \tag{16}$$

Since the watermark and the prompt guiding signal are interdependent, we employ an alternating optimization method, in which we iteratively optimize one while fixing the other. More details about the watermark design and optimization algorithm are presented in Appendix A.

## 3.3 Finding keypoints for implantation

The selection of the optimal stage for watermark embedding within the diffusion process is crucial for achieving both high image fidelity and semantic consistency with the input text prompt. We delve into the sensitivity of the predicted noise to frequency domain disturbances in different diffusion steps. According to classifier-free guidance method [27], the predicted noise in each step can be depicted as $Full = Uncondition + s \cdot (Condition - Uncondition)$. Condition and Uncondition are predicted noise with and without text conditions. Parameter $s$ is the scaling factor and the second term of the addition is called Guidance. Full noise is the final noise to be removed in the current step.

Figure 2(a) shows the evolution of mean values of various predicted noise terms throughout the generation process. We can find that after step 300, the slowdown in guidance rise indicates the completion of basic semantic formation and diminishing guidance influence. Additionally, Figure 2(b) presents how the predicted noise changes when perturbations are added at different timesteps. When the timestep is greater than 200, the frequency domain noise interferes with the generation process mainly by disrupting the guidance term. After 200 steps, the intrinsic unconditional term is more affected. We can conclude that early generation stages establish the foundation for image semantics and excessive intervention at this point can disrupt the intended image content. Conversely,

manipulating the final stages, dedicated to refining image details, may impede the model's capacity to recover from watermark-induced noise, ultimately compromising the final image quality.

Therefore, we strategically choose the watermark insertion point between steps 300 and 200. This stage offers the sweet spot: frequency perturbations have minimal impact on the mean of the predicted noise, allowing for watermark integration without sacrificing image quality and disruption to the core semantics.

## 3.4 Watermark validation

In the watermark verification phase, we reverse the diffusion process to get the state $\hat{x}_t^*$ at the watermark injection step. We extract the $\mathbb{M}$ region of $\hat{x}_t^*$' Fourier space and calculate its L1 distance from the implanted watermark $w_i$. However, the original prompt used for image generation is unknown during verification of online images. Similar to [44], we use the null-text prompt as the condition text embedding and set the guidance scale to 1.0. We also found unexpectedly that introducing the optimized prompt signal during inversion hinders the watermark recovery, which we aim to explore in future work. Our watermark verification requires a reversible generation process, making it compatible with any reversible samplers such as DPM-Solver [23], DPM-Solver++ [24], PNDM [21], and AMED-Solver [56].

# 4 Experiments

## 4.1 Experimental setting

**Model and dataset.** We conducted experiments on two distinct diffusion models operating in latent and image domains. For the latent diffusion model, we utilize the widely available Stable Diffusion-v2 [31] and the stable-diffusion-prompts dataset from Gustavosta [1]. We also test on a guided diffusion model [2] trained on the ImageNet [12], which operates directly on the pixel domain and can generate images of size $256 \times 256$ based on the category provided.

**Evaluation metrics.** To assess the effectiveness of ROBIN, we compute the Area Under the ROC Curve (AUC-ROC) based on the L1 distance to measure the effectiveness of watermark verification. Specifically, we compute AUC using 1,000 watermarked and 1,000 clean images. For the quality of watermarked images, we employ a suite of diverse metrics. We utilize classic measures like PSNR (Peak Signal-to-Noise Ratio), SSIM (Structural Similarity Index), and MSSIM (Multiscale SSIM) [42] to quantify the pixel-level differences between watermarked and original images. We employ the Fréchet Inception Distance (FID) [16] to evaluate the fidelity of the watermarked image distribution. We also leverage the CLIP score [29] to measure the alignment between generated images and their corresponding text prompts. More details are provided in Appendix B.1.

**Implementation details.** We utilize 50 steps of deterministic sampling for both models. Stable Diffusion employs the second-order multistep DPM-Solver algorithm [23] with a default guidance scale of 7.5. ImageNet diffusion model leverages the DDIM sampling algorithm [35]. We optimize the watermark and the hiding prompt using 50 generated images. The learning rates for the image watermark and prompt guidance are 0.8 and 5e-04, respectively, with a total of 1,000 optimization rounds. The default image watermark covers 70% of the image frequency domain. All experiments are conducted on an NVIDIA GeForce RTX 3090 GPU.

## 4.2 Effectiveness and robustness

We compare our method with five baselines: DwtDct [4], DwtDctSvd [26], RivaGAN [52], Stable Signature [14], and Tree-Ring watermarks [44]. To ensure the watermark's resilience in real-world scenarios, we delve into its robustness under various image transformations. These include Gaussian blur with radius 4, Gaussian noise with intensity 10%, jpeg compression with quality 25, color jitter with brightness 6, random rotation of 75 degrees, and random cropping of 75% and rescaling. These settings are strict for watermark verification because the image has been significantly altered. ROBIN is also evaluated under a combination of attacks where we randomly selected various combination of the six transformations. The processed samples are shown in Figure 6 in the Appendix.

Table 1: Comparison of AUC under different attacks and verification time for Stable Diffusion [31]. Clean represents the watermark verification on unmanipulated images. Avg is the average AUC across all attack cases. Time is the time required to validate the watermark for a single image.

| | Method | Clean | Blur | Noise | JPEG | Bright | Rotation | Crop | Avg | Time |
|---|---|---|---|---|---|---|---|---|---|---|
| Post-processing | DwtDct [4] | 0.974 | 0.503 | 0.293 | 0.492 | 0.519 | 0.596 | 0.640 | 0.574 | **0.056s** |
| | DwtDctSvd [26] | **1.000** | 0.979 | 0.706 | 0.753 | 0.517 | 0.431 | 0.511 | 0.702 | 0.233s |
| | RivaGAN [52] | 0.999 | 0.974 | 0.888 | 0.981 | 0.963 | 0.173 | 0.999 | 0.854 | 0.437s |
| In-processing | StableSig [14] | **1.000** | 0.565 | 0.731 | 0.989 | 0.976 | 0.658 | **1.000** | 0.845 | 0.112s |
| | Tree-Ring [44] | **1.000** | **0.999** | 0.944 | **0.999** | **0.983** | 0.935 | 0.961 | 0.975 | 2.599s |
| | ROBIN | **1.000** | **0.999** | **0.954** | **0.999** | 0.975 | **0.957** | 0.994 | **0.983** | 0.531s |

Table 2: Comparison of AUC under different attacks and verification time for Imagenet Diffusion [2]. Stable Signature is specifically designed for latent diffusion models and are incompatible with pixel-level ImageNet diffusion models

| | Method | Clean | Blur | Noise | JPEG | Bright | Rotation | Crop | Avg | Time |
|---|---|---|---|---|---|---|---|---|---|---|
| Post-processing | DwtDct [4] | 0.899 | 0.512 | 0.365 | 0.522 | 0.538 | 0.478 | 0.433 | 0.536 | **0.012s** |
| | DwtDctSvd [26] | **1.000** | 0.947 | 0.656 | 0.568 | 0.535 | 0.669 | 0.614 | 0.713 | 0.058s |
| | RivaGAN [52] | **1.000** | 0.988 | 0.962 | **0.978** | 0.924 | 0.321 | 0.999 | 0.882 | 0.109s |
| In-processing | Tree-Ring [44] | 0.999 | 0.975 | 0.979 | 0.940 | 0.861 | 0.975 | 0.994 | 0.966 | 3.963s |
| | ROBIN | **1.000** | **0.999** | **0.994** | 0.969 | **0.959** | **0.998** | **1.000** | **0.988** | 0.986s |

Table 3: AUC on different number of random attacks applied at the same time.

| Method | 1 | 2 | 3 | 4 | 5 | 6 |
|---|---|---|---|---|---|---|
| Tree-Ring | 0.969 | 0.809 | 0.699 | 0.520 | 0.546 | 0.509 |
| ROBIN | **0.973** | **0.814** | **0.759** | **0.579** | **0.558** | **0.556** |

**Robustness.** The comprehensive results of AUC comparison with baselines are presented in Table 1 and Table 2. While most methods (except DwtDct) perform well for watermark verification in the absence of attacks, their accuracy degrades with strong image manipulations. Traditional frequency-domain methods show significant vulnerability. RivaGAN falters with image rotations, and Stable Signature exhibits sensitivity to blur, noise, and rotation. The Tree-Ring watermark displays better robustness due to its pattern design but remains less resilient than ROBIN.

The performance of watermark verification for Stable Diffusion under different numbers of simultaneous attacks is shown in Table 3. Note that due to the inherent potency of the individual attacks, their combination leads to significant image quality deterioration. The resulting images are presented in Figure 7. But ROBIN still demonstrates superior robustness compared to the state-of-the-art method Tree-Ring in such challenging scenarios.

The robustness of ROBIN on the one hand comes from the introduction of an explicit hiding process, we can implant a stronger watermark. Furthermore, fewer inversion steps during verification compared to Tree-Ring watermarks also mitigate the accumulation of DDIM inversion errors, further enhancing accuracy. The evaluation of ROBIN under more attacks is presented in Appendix C.2.

**Time cost.** The time cost of watermark verification associated with different watermarking schemes is presented in the last column of Table 1 and Table 2. The simple DwtDct method demonstrates the fastest performance, achieving a validation time of less than 0.1s. DwtDctSvd exhibits a $4\times$ slowdown compared to DwtDct, while RivaGAN is $10\times$ slower. StableSig decodes the watermark directly from the image, but it requires fine-tuning the model. The verification of Tree-Ring watermarks necessitates reversing the entire generation process, resulting in significant time costs. ROBIN requires reversing only a limited number of generation steps, resulting in consumption times of 0.531s and 0.986s for the two models, which are considerably lower compared to the Tree-Ring watermark. More experimental results are presented in Appendix C.1.

### 4.3 Quality of watermarked image

Traditional post-hoc watermarking methods introduce subtle visual distortions into the generated images. In contrast, the objective of ROBIN aligns with the Tree-Ring in constructing a "content

Table 4: Quality of generated images. PSNR, SSIM and MSSIM measure the similarity between the watermarked and unwatermarked images. CLIP evaluates how well the watermarked image aligns with the user-provided textual description. FID measures the distribution similarity between the watermarked dataset and a random dataset of real images. The subscripts indicate the standard deviation of five independent experimental runs, each initialized with a different random seed.

| Model | Method | PSNR ↑ | SSIM ↑ | MSSIM ↑ | CLIP ↑ | FID ↓ |
|---|---|---|---|---|---|---|
| Stable Diffusion [31] | W/o watermark | $\infty$ | 1.000 | 1.000 | 0.403 | 25.53 |
| | Tree-Ring [44] | $15.37_{.07}$ | $0.568_{.003}$ | $0.626_{.005}$ | $0.364_{.00}$ | $\mathbf{25.93_{.13}}$ |
| | ROBIN | $\mathbf{24.03_{.04}}$ | $\mathbf{0.768_{.000}}$ | $\mathbf{0.881_{.001}}$ | $\mathbf{0.396_{.00}}$ | $26.86_{.09}$ |
| ImageNet Diffusion [2] | W/o watermark | $\infty$ | 1.000 | 1.000 | 0.271 | 16.25 |
| | Tree-Ring [44] | $15.68_{.03}$ | $0.663_{.002}$ | $0.607_{.001}$ | $0.267_{.00}$ | $\mathbf{17.68_{.16}}$ |
| | ROBIN | $\mathbf{24.98_{.02}}$ | $\mathbf{0.875_{.000}}$ | $\mathbf{0.872_{.000}}$ | $\mathbf{0.275_{.00}}$ | $18.26_{.13}$ |

**W/o Watermark**   **Tree-Ring**   **ROBIN**

"Young, curly haired redhead girl in a dark medieval inn"

"Full body portrait of white haired girl in spider man suit"

"Cloudscape, nebula gasses in the background, fantasy magic angel"

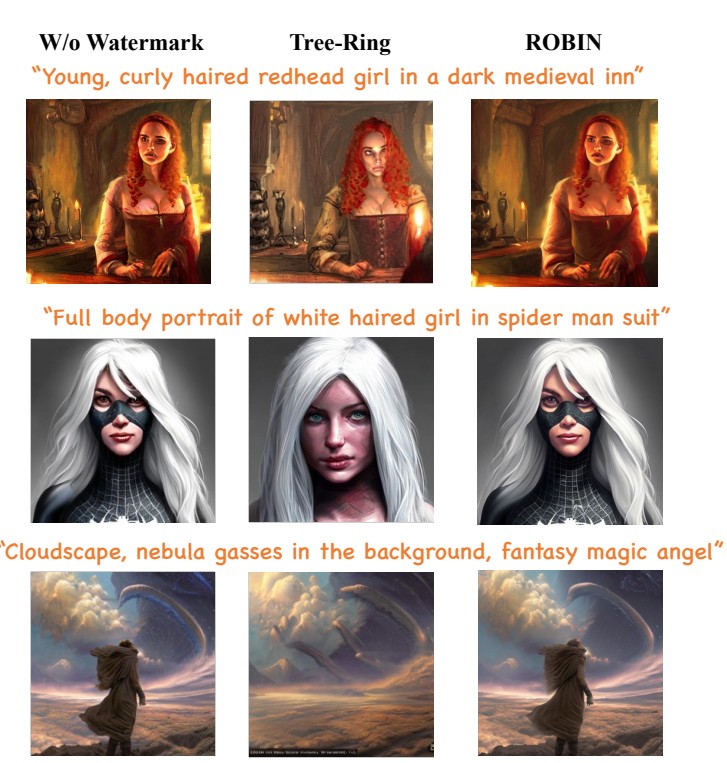

Figure 3: The generated images with Tree-Ring and ROBIN watermarks.

watermark": seamlessly embedding the watermark within the image content without altering its semantics. Due to this fundamental shift in watermarking philosophy, we only compare the image quality with Tree-Ring watermarks.

The Tree-Ring approach aims to find another watermarked image that aligns with the text prompt, even if it differs from the original image. However, it is more akin to random semantic modifications and does not guarantee the same level of text alignment as the original generation. Figure 3 shows that the Tree-Ring approach significantly alters the generated image's semantics, sometimes even failing to fulfill the text prompt's intent. This occurs because it disrupts the essential Gaussian characteristics of the initial noise, hindering the generation process. In contrast, ROBIN excels at preserving the overall image content and semantic structure, providing a better lower bound for faithfulness by preserving the original semantics. Table 4 provides the quantitative results. ROBIN demonstrates significant improvements in PSNR, SSIM, MSSSIM, and CLIP score, while a slight increase in FID is observed. This is because the position of the watermark implanted in our scheme is at a later stage of generation, resulting in a slightly greater influence on the overall generation distribution. This implies a negligible trade-off for achieving a strong watermark with minimal degradation of the overall quality of the generated image.

Table 5: Watermark accuracy and image quality under different settings. (1) random watermarks $w_i$, (2) random watermarks with prompt signal $w_p$ for hiding, (3)-(5) different loss functions for optimizing $w_i$ and $w_p$, (6) full loss function for optimizing both $w_i$ and $w_p$.

| ID | Watermark | | Loss function | | | AUC↑ | | Image quality | | | |
|----|-----------|-----------|-------------|--------------|-----------|-------|-------------|--------|-------|-------|------|
| | Image $w_i$ | Prompt $w_p$ | $\ell_{ret}$ | $\ell_{cons}$ | $\|w_i\|$ | Clean | Adversarial | PSNR ↑ | SSIM↑ | CLIP↑ | FID↓ |
| (1) | Random | None | | | | 1.00 | 0.903 | 20.11 | 0.68 | 0.39 | 29.21 |
| (2) | Random | Optimized | ✓ | ✓ | | 1.00 | 0.901 | 21.70 | 0.70 | 0.39 | 27.77 |
| (3) | Optimized | Optimized | | ✓ | ✓ | 1.00 | 0.988 | 18.95 | 0.48 | 0.30 | 32.18 |
| (4) | Optimized | Optimized | ✓ | | ✓ | 1.00 | 0.970 | 23.91 | 0.76 | 0.40 | 26.68 |
| (5) | Optimized | Optimized | ✓ | ✓ | | 1.00 | 0.966 | 24.19 | 0.77 | 0.40 | 26.93 |
| (6) | Optimized | Optimized | ✓ | ✓ | ✓ | 1.00 | 0.983 | 24.03 | 0.77 | 0.40 | 26.86 |

## 4.4 Ablation study

To gain further insights into the effectiveness of ROBIN, we conduct an ablation study, exploring the influence of different design choices. We additionally introduce the Mean Squared Error of Watermark (MSE) to represent the verification accuracy in some settings where the AUC is always equal to 1. It is calculated as the mean of L1 distance between the extracted and original watermark.

**Setting variations.** To explore the individual contributions of various components in our scheme, we conduct a series of experiments presented in Table 5. Experiments in Settings 1 and 2 demonstrate that the introduction of prompt-based watermark hiding signals improves image quality, as evidenced by a 1.6 increase in PSNR and a 1.44 decrease in FID score compared to Setting 1. Setting 3 emphasizes the importance of the $\ell_{ret}$ in controlling watermark strength. Without $\ell_{ret}$, ROBIN prioritizes creating a highly robust watermark, leading to significant image distortion (PSNR: 18.95, SSIM: 0.48). Setting 4 presents that removing $\ell_{cons}$ allows for stronger prompt guidance, but this results in increased DDIM inversion loss and a decrease of 0.13 in adversarial AUC. Setting 5 prioritizes minimal impact on the generated image by weakening the watermark. This approach leads to poorer watermark robustness and a decrease of 0.017 in adversarial AUC. Experiments under Settings 2 and 6 demonstrate that in the presence of the hiding prompt signal, the image watermark can be optimized to achieve stronger robustness while maintaining invisibility.

**Point of implantation.** We evaluate the impact of implanting the watermark at different stages in the diffusion process. The results are presented in Figure 4. Watermark verification accuracy improves with later implantation due to fewer DDIM inversion steps and reduced information loss. Early implantation, while initially maintaining image quality (low FID), can significantly change the image content (low SSIM/PSNR) by disrupting semantic formation. Conversely, late implantation may leave the watermark visible due to insufficient space for hiding, leading to high FID and deviation from the original image (low SSIM). This empowers us to pinpoint the optimal embedding stage (steps 15-10) for balancing visual quality and semantic preservation.

**Watermark strength.** We also verify the influence of different watermarking strengths and the results are shown in Figure 4. Higher watermark strength (proportional coverage in the frequency domain) generally benefits verification accuracy, as the watermark becomes more prominent. The CLIP score and FID remain stable due to strategic embedding and guided hiding. Traditional metrics (SSIM, PSNR) decrease with stronger watermarks due to increased content modification. The watermarked images under different strengths are shown in Figure 5. Compared to Tree-Ring, the quality of generated images with ROBIN watermarks is less sensitive to watermark strength. More qualitative results are presented in Appendix C.5.

## 5 Conclusion & Discussion

This paper proposes a novel watermarking method for the diffusion model, which embeds a watermark in the intermediate diffusion state and guides the model to conceal the watermark. By explicitly introducing the active hiding process, we can implant stronger watermarks without compromising image quality. We believe this method holds promise for expanding the possibilities of reliable watermarking in diffusion models.

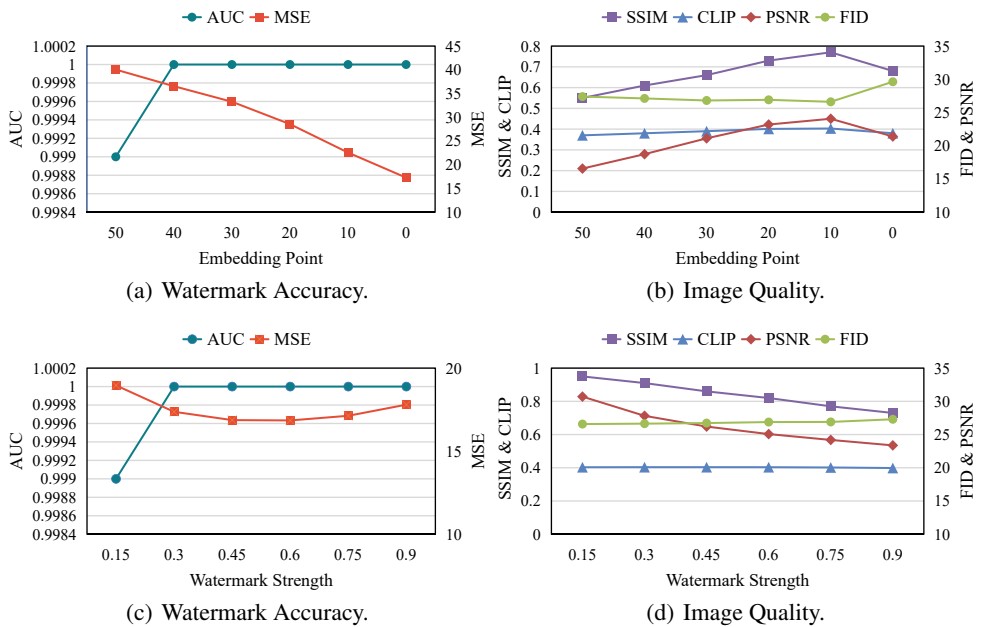

(a) Watermark Accuracy.  (b) Image Quality.

(c) Watermark Accuracy.  (d) Image Quality.

Figure 4: Ablation experiments on embedding point and watermark strength.

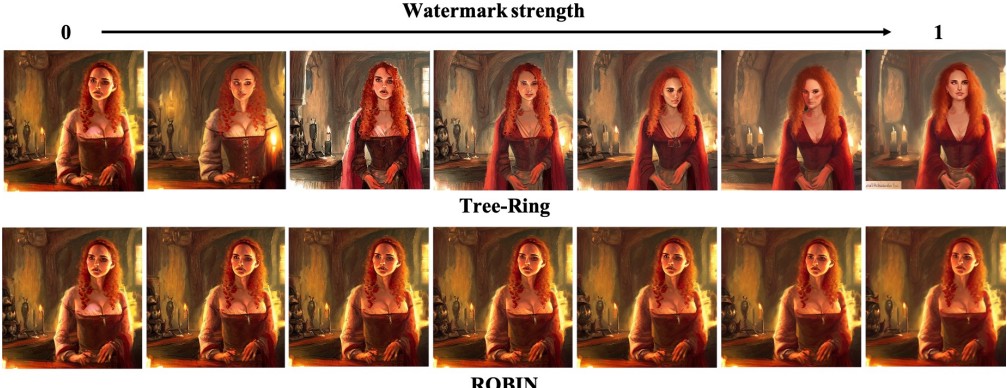

Figure 5: Generated images under different watermark strengths. The top row is the result of the Tree-Ring scheme and the bottom row is the result of ROBIN.

**Limitations.** The verification of ROBIN watermarks relies on the reversible generation process, future advancements enabling the reversibility of other sampling algorithms would broaden the application of our method. Additionally, the inherent information loss during DDIM inversion can be reduced by exploring generative trajectories that can be reversed exactly [28, 18, 40, 51].

**Social impact.** Our ROBIN scheme, as a watermarking method, can help creators establish ownership and discourage unauthorized use. Furthermore, ROBIN watermarks can be implanted in a one-shot manner without retraining the whole model, making it applicable to different diffusion-based text-to-image models.

# Acknowledgment

This work was partially supported by the National Natural Science Foundation of China under grants 62372341, U20B2049, and U21B2018.

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

# Appendix

## A  Scheme details

### A.1  Image watermark design

To make the watermark less visible and more resistant to alterations, we embed the watermark into the frequency domain of the selected diffusion latent. Frequency domain watermarks are proven to be robust against common manipulations like cropping and compression and resilient against geometric distortions, such as scaling and rotation. Draw inspiration from Tree-Ring watermarks, we use a radiating watermark pattern, where the watermark information within each frequency band holds equal values. This design choice enhances the watermark's robustness against image rotations. Specifically, after each optimization round of the image watermark, the values within a specific frequency band are averaged. This averaged pattern is then used for further prompt signal optimization and another round of adversarial optimization.

### A.2  Prompt signal design

Our scheme is based on the classifier-free guidance technique, where the generation relies on both unconditional and conditional predictions. The predicted noise of $x_t$ at step $t$ is defined as:

$$\tilde{\epsilon}_\theta(x_t, t, \psi(p)) = \eta \cdot \epsilon_\theta(x_t, t, \psi(p)) + (1 - \eta) \cdot \epsilon_\theta(x_t, t, \psi(\emptyset)), \tag{17}$$

where $\eta$ is the guidance scale parameter and $p$ is the input text condition. In this way, the model can maintain its original ability to remove noise and the new function of generating specific content.

### A.3  Optimization algorithm design

The details of the adversarial optimization algorithm are presented in Algorithm 1. Initially, the image watermark $w_i$ is randomly sampled, and guidance $w_p$ is set to NULL (representing no text prompt). In each round, we randomly select a generated sample $x_0$ and obtain the noise representation $x_t$ at the watermark embedding point. Then both $w_i$ and $w_p$ are optimized alternatively in an adversarial manner. We experimentally set the hyperparameters $\alpha$ as 1.0, $\beta$ as 1.0, and $\lambda$ as 0.005.

---

**Algorithm 1** Adversarial Optimization Algorithm

---

**Input:** Dataset $\mathcal{X}, \mathcal{P}$; max epoch $N$; hyper-parameters $\alpha, \beta, \lambda$; watermark mask $\mathbb{M}$
**Output:** Optimized watermark pair $w_i, w_p$;
1:  Initialization
    $w_i^0 \leftarrow \texttt{rand\_init}(w_i)$ ;
    $w_p^0 \leftarrow \psi(\texttt{NULL})$ ;
    $k \leftarrow 0$ ;
2:  **while** not converged yet **do**
3:      // get sample
    $x_0, p \sim \mathcal{X}, \mathcal{P}$
4:      //get $x_t$ from $x_0$
    $x_t \leftarrow \sqrt{\bar{\alpha}_t}x_0 + \sqrt{1 - \bar{\alpha}_t}\epsilon, \epsilon \sim \mathcal{N}(0, \mathbf{I}), t \sim \texttt{Uniform}(200, 300)$
5:      // image watermark optimization
    $w_i^{k+1} = \underset{w_i}{\arg\min}(\alpha\ell_{ret} + \beta\ell_{cons} - \lambda\|w_i^k\|)$
6:      // prompt guidance optimization
    $w_p^{k+1} = \underset{w_p}{\arg\min}(\alpha\ell_{ret} + \beta\ell_{cons})$
    $k \leftarrow k + 1$
7:  **end while**
8:  **return** $(w_i^k, w_p^k)$

---

Table 6: Comparision of time cost (s) of different watermarking methods.

| Method | | Stable Diffusion [31] (512×512) | | Imagenet Diffusion [2] (256×256) | |
| --- | --- | --- | --- | --- | --- |
| | | Generation | Validation | Generation | Validation |
| W/o Watermark | | 2.614 | - | 3.479 | - |
| Post-processing | DwtDct [4] | 2.681 | 0.056 | 3.492 | 0.012 |
| | DwtDctSvd [26] | 2.749 | 0.233 | 3.511 | 0.058 |
| | RivaGAN [52] | 3.342 | 0.437 | 3.661 | 0.109 |
| In-processing | StableSig [14] | 2.614 | 0.112 | - | - |
| | Tree-Ring [44] | 2.617 | 2.599 | 3.482 | 3.963 |
| | ROBIN | 2.682 | 0.531 | 3.592 | 0.986 |

## B    Implementation details

### B.1    Details about evaluation metric

**Setting for AUC computing.**    The AUC-ROC (Area Under the ROC Cure) metric is a statistical measure used to evaluate the performance of a binary classification problem, which is watermarked or not here. The ROC Curve is created by plotting the fraction of true positive results against the fraction of false positive results at various threshold settings. The AUC summarizes the overall performance across all possible thresholds. A higher AUC value means the test is more accurate in making this distinction. And an AUC of 1.0 represents perfect discrimination. For both our method and Tree-Ring watermarking, we compare the extracted image watermarks using the L1 distance. For the other three steganography-based methods, we utilize the Hamming distance between the implanted and decoded binary sequences, as these methods typically operate on binary data representations.

**Setting for FID computing.**    For Stable Diffusion, we generate 5,000 watermarked images and calculate FID against the MS-COCO-2017 dataset [20]. For the ImageNet Diffusion model, we calculate FID using 10,000 watermarked images against the ImageNet-1K training dataset [11].

**Setting for CLIP computing.**    For both models, we test 1,000 images using the OpenCLIP-ViT model [6]. For Stable Diffusion, we work with the ground-truth text prompts, while for the ImageNet model, we construct prompts like "a photo of x", where "x" is the category of the generated image.

**About pixel-level metrics.**    The content watermarking scheme of ROBIN doesn't aim for exact replication of the original image. Instead, it strives for a visually similar "alternative generation" that maintains both image quality and semantic integrity. While traditional watermarking schemes utilized metrics like PSNR/SSIM to assess image distortion introduced by the watermark (treated as an additional signal), we utilize them in this research as supplementary indicators to reflect the degree of semantic preservation within the watermarked image. Essentially, the higher the similarity between the watermarked and original image, the less semantic impact the watermark has introduced.

## C    More experimental results

### C.1    Time overhead

We evaluate the time cost associated with different watermarking schemes. The results are presented in Table 6. Traditional post-processing methods exhibit similar time requirements for watermark addition and verification. The simple DwtDct method demonstrates the fastest performance, achieving both addition and validation times of less than 0.1s. DwtDctSvd exhibits a 3× slowdown compared to DwtDct, while RivaGAN is 10× slower. Notably, the runtime of these methods is heavily influenced by the input image size. For in-processing watermarking, StableSig directly fine-tunes the model, incurring no additional time overhead during the generation process. The Tree-Ring method introduces minimal impact (0.003s) on generation time by solely modifying the initial random vector. However, verification necessitates reversing the entire generation process, resulting in significant time consumption (2.6s for Stable Diffusion and 3.9s for Imagenet Diffusion). ROBIN employs a one-shot approach for watermark embedding during the intermediate diffusion stage. The impact

Table 7: Watermark verification (AUC) under reconstruction attack.

| Method | VAE-Bmshj2018 | VAE-Cheng2020 | Diffusion model |
|---|---|---|---|
| Tree-Ring | 0.992 | 0.993 | 0.996 |
| ROBIN | **0.998** | **0.999** | **0.997** |

Table 8: Watermark verification (AUC) on noise-to-image generation.

| Diffusion Type | Clean | Blur | Noise | JPEG | Bright | Rotation | Crop | Avg |
|---|---|---|---|---|---|---|---|---|
| Noise-to-Image | 1.0 | 0.996 | 0.997 | 1.0 | 0.963 | 0.999 | 1.0 | 0.993 |

Table 9: Watermark verification (AUC) on different optimization settings.

| Alignment | Clean | Blur | Noise | JPEG | Bright | Rotation | Crop | Avg |
|---|---|---|---|---|---|---|---|---|
| Latent-level | 1.000 | 0.999 | 0.940 | 0.999 | 0.974 | 0.927 | 0.994 | 0.972 |
| Pixel-level (ROBIN) | 1.000 | 0.999 | 0.954 | 0.999 | 0.975 | 0.957 | 0.994 | 0.983 |

on generation time arises from the introduction of additional guidance calculations, resulting in a minimal overhead of 0.068s and 0.113s, which is negligible compared to the generation time of 2.614s and 3.479s for the two models. Verification of ROBIN watermarks requires reversing only a limited number of generation steps, resulting in consumption times of 0.531s and 0.986s for the two models, which are considerably lower compared to the Tree-Ring watermark.

## C.2 Reconstruction attacks

We evaluate the performance of ROBIN under different variants of reconstruction attacks [54]. As shown in Table 7, ROBIN consistently exhibits stronger robustness under these adversarial conditions.

## C.3 Application to noise-to-image models

ROBIN can also be applied to noise-to-image generation models, as it does not rely on the original text prompt input. Given that large-scale pretrained diffusion models are typically conditional generative models, we chose to use the unconditional capability of Stable Diffusion to simulate the noise-to-image generation process for this evaluation.

We evaluate ROBIN on the unconditional generation of Stable Diffusion, where the original text is set to NULL (representing no text prompt). In this setup, the image is generated unconditionally before the watermark injection point. After that, we still utilize our watermarking hiding prompt embedding to guide the generation process and actively erase the watermark. The results in Table 8 indicate that ROBIN can still function well in noise-to-image generation.

## C.4 Pixel-level optimization

In our scheme, the watermark is embedded in the latent space while the loss function is calculated at the pixel level. We believe that this approach, which combines pixel-level alignment with latent space optimization, is beneficial for improving robustness.

This is because different latent representations can map to similar pixel-level expressions, allowing us to find a latent code that maps to visually the same image but also contains robust watermarks. This provides more opportunities to embed strong and robust watermark signals without introducing noticeable visual artifacts. The benefits of this optimization method are evident when we actively aim for concealment, a feature not supported by other watermarking methods.

To further validate our approach, we also test a variant of the ROBIN scheme where the loss function is computed at the latent level rather than the pixel level. The results presented in Table 9 demonstrate that latent-level alignment slightly decreases the robustness of the watermark, thereby underscoring the effectiveness of our pixel-level alignment strategy.

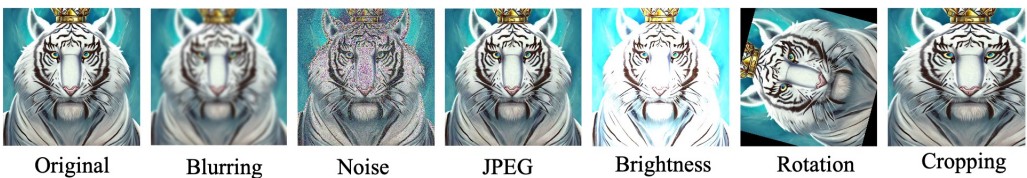

| Original | Blurring | Noise | JPEG | Brightness | Rotation | Cropping |

Figure 6: Samples under different attacks.

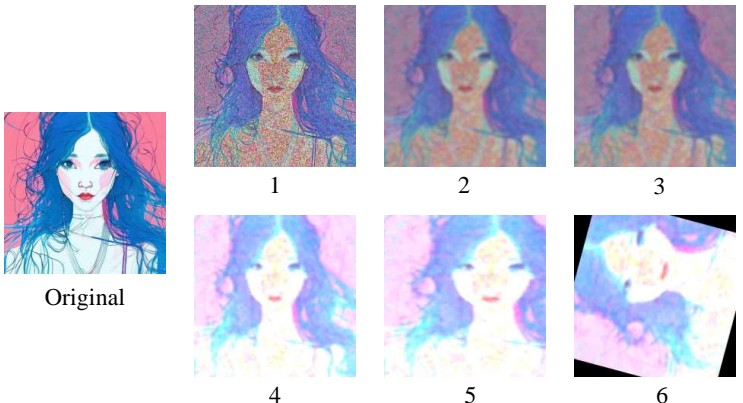

Figure 7: Samples under different number of attacks applied at the same time. The sequence of attacks performed on the above images is Gaussian blur with radius 4, JEPG compression with quality 25, color jitter with brightness 6, random cropping of 75%, and random rotation of 75 degrees.

## C.5  More qualitative results

**W/o Watermark**   **Tree-Ring**   **ROBIN**

"Glowing cracks, elven princess, meditating, lotus pose, blossoming"

"Full samurai ninja armor, spiderman, fantastic details full face"

"Anime as Margot Robbie cute-fine-face, surprised realistic shaded face"

"Oil painting portrait of a young black woman with long hair in a white dress"

"A knitted Capybara wearing stylish sunglasses and dressed in a beanie cap"

"Dark fantasy, evil magician portrait, dark surrealist"

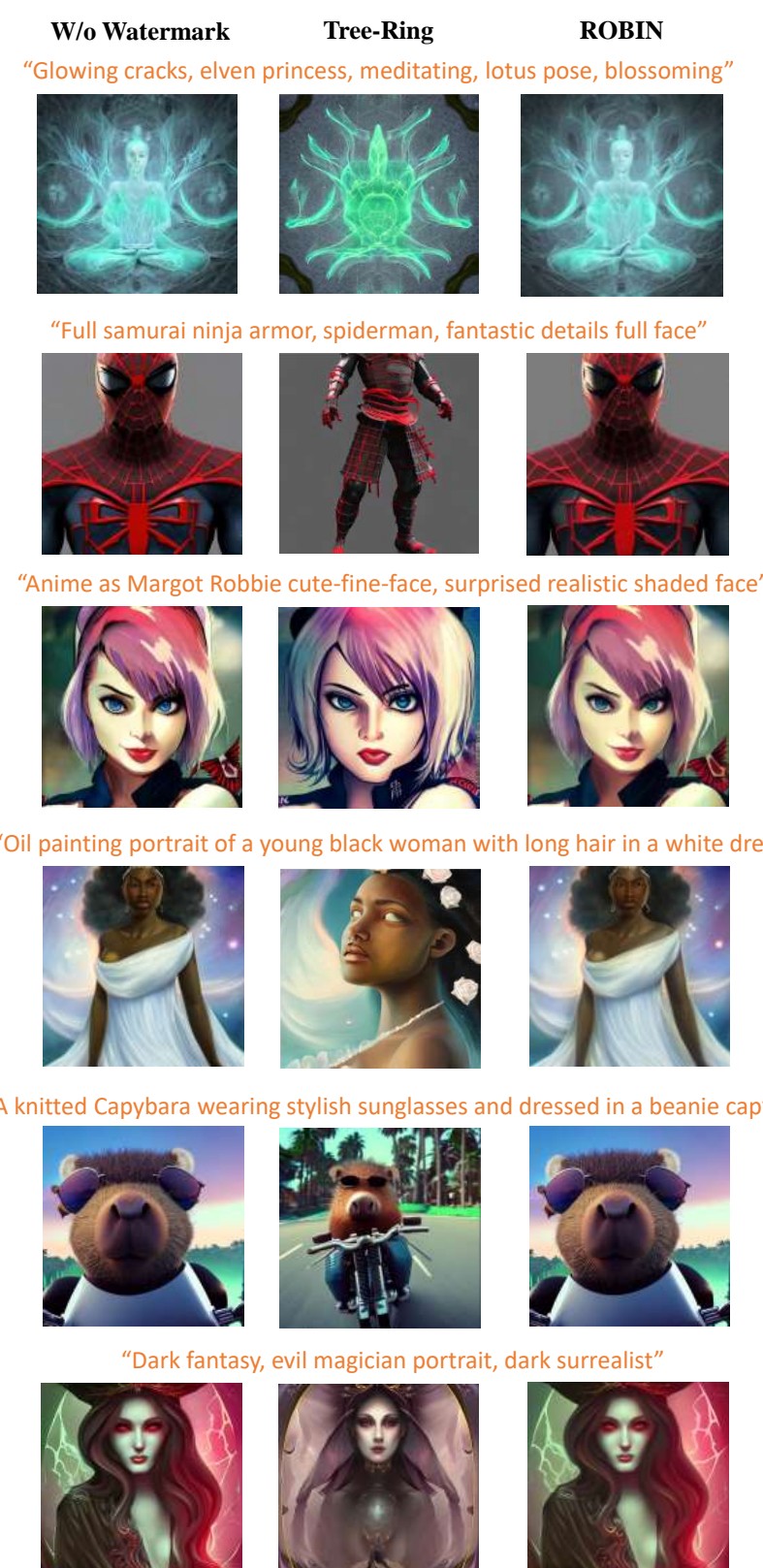

Figure 8: More qualitative comparison results with Tree-Ring watermarks for Stable Diffusion.

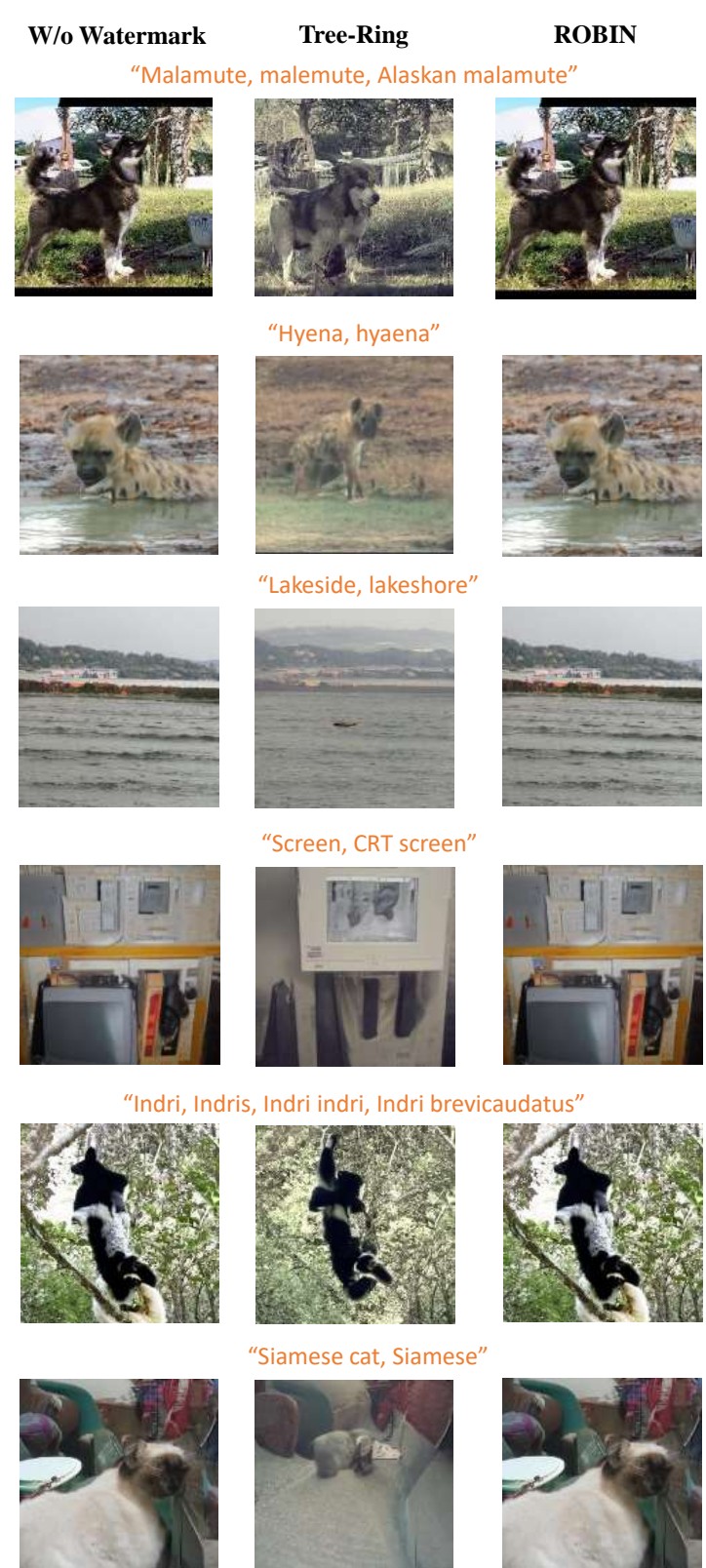

Figure 9: More qualitative comparison results with Tree-Ring watermarks for the ImageNet Diffusion model.

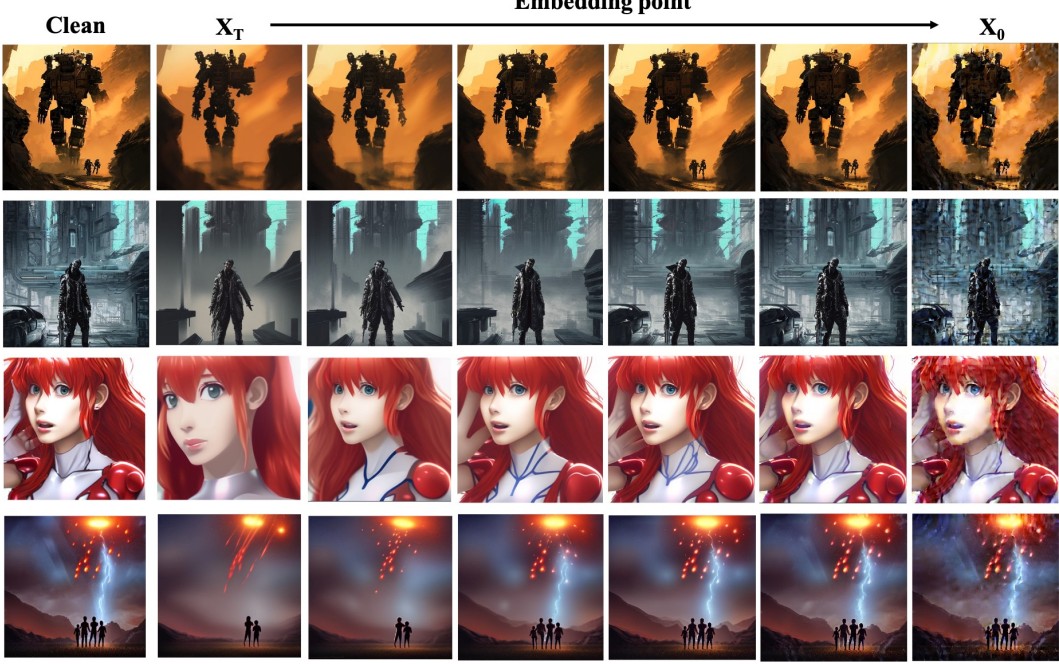

Figure 10: Generated images with the watermark embedded at different diffusion stages. Clean represents the images that are generated without watermarking. $X_T$ means the watermark is embedded into the initial noise, and $X_0$ means the watermark is implanted in the final generated image.

