# OpenReview forum: "ROBIN: Robust and Invisible Watermarks for Diffusion Models with Adversarial Optimization"
_NeurIPS.cc/2024/Conference — NeurIPS 2024 poster_

### Official Review · Reviewer_4gSJ · 2024-07-12

**Soundness:** 3
**Presentation:** 3
**Contribution:** 3
**Rating:** 6
**Confidence:** 3

**Summary:**

This paper presents a novel watermarking technique for diffusion models that is robust against input transformations and invisible to the human eye. Unlike existing methods that rely on post-processing or perturbing the initial noise, this technique actively injects the watermark signal during the intermediate diffusion process. The method employs adversarial optimization to maximize watermark strength while minimizing the difference between the watermarked and original outputs. It also identifies optimal keypoints for embedding watermarks without compromising image quality.
During verification, the technique measures the distance between the holdout watermark and the reconstructed watermark from the intermediate stage; a small distance confirms the presence of the injected watermark.
The evaluation part considers six types of attacks on watermarking, with results showing that the proposed method outperforms existing baselines in terms of robustness. Additionally, numerical and perceptual assessments indicate that the proposed method introduces less distortion in image quality compared to baselines. Several ablation studies validate the design choices of the proposed method.

**Strengths:**

1. The design incorporates an intriguing adversarial optimization and prompt embedding optimization strategy.
2. Well-written and easy to follow.
3. Results are promising, validated through thorough testing.

**Weaknesses:**

1. The evaluation could be enhanced by including more strong attack scenarios.
2. Certain sections of the manuscript would benefit from further detailed explanations.

**Questions:**

(1) While the robustness demonstrated in Table 1 is commendable, could the method maintain its efficacy against a combination of attacks, such as Blur, Rotation, and Crop? Additionally, considering [1] highlights the effectiveness of reconstruction attacks, a discussion on these would be insightful.

(2) Is the proposed watermarking technique applicable to noise-to-image diffusion models? Further discussion on this application would be valuable.

(3) Equation (10) suggests that the watermarking operates at a pixel level; how does this contribute to its robustness against numerous attacks? It would be beneficial for the readers if the authors could provide deeper insights into the mechanisms that ensure robustness, beyond just presenting successful outcomes.

(4) The abstract mentions that existing watermarking methods are passive, whereas the proposed method is active. Could the authors clarify this distinction? The current explanation within the manuscript does not fully convey the implications of this difference.

Reference:
[1] Zhao, Xuandong, et al. Invisible Image Watermarks Are Provably Removable Using Generative AI. ArXiv:2306.01953 (2023).

**Limitations:**

This paper has included the potential limitation.

---

> ### Author Rebuttal · Authors · 2024-08-07
>
> > Q1. Could the method maintain its efficacy against a combination of attacks and reconstruction attacks?
>
>
> As suggested by the reviewer, we have evaluated ROBIN under both conbination attacks and reconstruction attacks.
>
> (1) We randomly selected various combinations of the six attacks outlined in this paper. The performance of watermark verification under different numbers of simultaneous attacks is shown in the table below.
>
> Note that due to the inherent potency of the individual attacks, their combination leads to significant image quality deterioration. The resulting images are included in the attached rebuttal pdf. These images are no longer suitable for watermark detection as their integrity has been severely compromised. But ROBIN still demonstrates superior robustness compared to the state-of-the-art method Tree-Ring in such challenging scenarios.
>
> Table 1. Watermark verification (AUC) on different number of random attacks applied at the same time.
> | Method | 1 | 2 | 3 | 4 | 5 | 6 |
> | :---- | :---- | :---- | :---- | :---- | :---- | :---- |
> | Tree-Ring | 0.969 | 0.809 | 0.699 | 0.520 | 0.546 | 0.509 |
> | ROBIN | **0.973** | **0.814**| **0.759**| **0.579** | **0.558** | **0.556** |
>
> (2) We evaluate the performance of ROBIN under different variants of reconstruction attacks [a]. As shown in the table below, ROBIN consistently exhibits stronger robustness under these adversarial conditions.
>
> Table 2. Watermark verification (AUC) under reconstruction attack.
> | Method | VAE-Bmshj2018 | VAE-Cheng2020 | Diffusion model |
> | :---- | :---- | :---- | :---- |
> | Tree-Ring | 0.992 | 0.993 | 0.996 |
> | ROBIN | **0.998** | **0.999** | **0.997** |
>
> [a] Invisible Image Watermarks Are Provably Removable Using Generative AI. ArXiv:2306.01953 (2023).
>
> > Q2. Is the proposed watermarking technique applicable to noise-to-image diffusion models?
>
> ROBIN can indeed be applied to noise-to-image generation models, as our scheme does not rely on the original text prompt input. Given that large-scale pretrained diffusion models are typically conditional generative models, we chose to use the unconditional capability of Stable Diffusion to simulate the noise-to-image generation process for this evaluation.
>
> We evaluate ROBIN on the unconditional generation of Stable Diffusion, where the original text is set to NULL (representing no text prompt).
> In this setup, the image is generated unconditionally before the watermark injection point. After that, we still utilize our watermarking hiding prompt embedding $w_p$ to guide the generation process and actively erase the watermark.
> The results in the table below indicate that ROBIN can still function well in noise-to-image generation.
>
> Table 3. Watermark verification (AUC) on noise-to-image generation.
> | Diffusion Type | Clean | Blur | Noise | JPEG | Bright | Rotation | Crop | Avg |
> | :---- | :---- | :---- | :---- | :---- | :---- | :---- | :---- | :---- |
> | Noise-to-Image | 1.0 | 0.996 | 0.997 | 1.0 | 0.963 | 0.999 | 1.0 | 0.993 |
>
> > Q3. Provide more insights about how the watermarking at a pixel level contributes to the robustness.
>
> (1) In our scheme, the watermark is embedded in the latent space while the loss function is calculated at the pixel level (Equation 10 in the original manuscript).
> We believe that this approach, which combines pixel-level alignment with latent space optimization, is beneficial for improving robustness.
>
> (2) This is because different latent representations can map to similar pixel-level expressions, **allowing us to find a latent code that maps to visually the same image but also contains robust watermarks**.
> This provides more opportunities to embed strong and robust watermark signals without introducing noticeable visual artifacts.
> The benefits of this optimization method are evident when we actively aim for concealment, a feature not supported by other watermarking methods.
>
> (3) To further validate our approach, we also tested a variant of the ROBIN scheme where the loss function is computed at the latent level rather than the pixel level. The results presented in the table below demonstrate that latent-level alignment slightly decreases the robustness of the watermark, thereby underscoring the effectiveness of our pixel-level alignment strategy.
>
> Table 4. Watermark verification (AUC) on different optimization settings.
> | Alignment | Clean | Blur | Noise | JPEG | Bright | Rotation | Crop | Avg |
> | :---- | :---- | :---- | :---- | :---- | :---- | :---- | :---- | :---- |
> | Latent-level | 1.000 | 0.999 | 0.940 | 0.999 | 0.974 | 0.927 | 0.994 | 0.972 |
> | Pixel-level (ROBIN) | 1.000 | 0.999 | 0.954 | 0.999 | 0.975 | 0.957 | 0.994 | 0.983 |
>
> > Q4. Clarify more about the distinction between the passive nature of existing methods and the active nature of the proposed method.
>
>
> We appreciate the reviewers' comments and would like to clarify this point in more detail.
>
> (1) Existing methods achieve concealment passively because they don't take invisibility into consideration during watermark implantation and don't take actions to achieve this goal actively. Users are thus compelled to manually and empirically reduce the watermark strength to achieve stealthiness, often leading to a weak watermark signal and decreased robustness.
>
>
> (2) In our method, we prioritize invisibility as a goal for watermark embedding and design our approach specifically to achieve this goal.
> We design a prompt embedding guidance to achieve invisibility, and our optimization of prompt embeddings guides the model to visually conceal the watermark. Consequently, our method can maximize watermark strength while minimizing visual artifacts.

---

> > ### Comment · Reviewer_4gSJ · 2024-08-08
> >
> > I appreciate the efforts made by the authors in addressing my concerns and questions!
> >
> > I'm glad to see the additional experimental results which provide a lot of insights.
> >
> > Hence, I increase my rating and hope the authors could properly include these results and discussions into the final version.

---

> > > ### Author Response · Authors · 2024-08-13
> > > **Thank you for recognizing our responses!**
> > >
> > > Dear Reviewer 4gSJ, thanks for recognizing our work. We are happy that our response has addressed your concerns.
> > >
> > > We will definitely include these discussions and additional experiments in our final version.
> > >
> > > Thank you for helping to improve our work again!

---

### Official Review · Reviewer_duRp · 2024-07-12

**Soundness:** 3
**Presentation:** 3
**Contribution:** 3
**Rating:** 6
**Confidence:** 4

**Summary:**

This paper discuss ROBIN, robust and invisible watermarks. The method proposes to embed the watermark during intermediate steps of the sampling process of diffusion model, by optimized prompt guidance signal w_p, the model was able to embed the invisible watermark into the generated content without losing the image quality while maintaining high robustness. The watermarked image can later be decoded through DDIM inversion.

**Strengths:**

This paper discusses a significant problem of improving the trade off between image quality and watermarking robustness. The idea of optimizing the guiding prompt in at a later stage of the diffusion steps is interesting and novel.

**Weaknesses:**

The main issue of this paper is the presentation of methodology and empirical results. Several key aspects are unclear.
1. From the main context, it is unclear how w_p and w_i are optimized, although equations 13 and 14 do provide a loss function to w_p and w_i, line 5 and line 6 in algorithm 1 provide different formulations than simply minimizing equations 13 and 14. Additionally, in equations 8 and 11, the author mentioned $\epsilon_\theta(x_t^*,t,\psi(p), w_p)$, which is not defined until Appendix A. I personally suggest the author to move the important part of Appendix A to the main text of the paper to enhance readability and reduce confusion since they are pivotal to understanding the methodology.
 2. Another layer of confusion comes from the empirical results, for example, in caption of figure 2, it mentions that "Guidance is the guide signal calculated from the Condition and Uncondition". I personally cannot understand how the author calculated the guidance, so I unfortunately fail to understand the meaning of this figure. Additionally for figure 2a, the red "Uncondition" curve is not visible in the figure, if the curve completely overlaps with another curve, it will be helpful if the author could indicate that in the text, or change the way of presentation for figure 2a.
3. In table 3, the author did not explicitly explain the meaning of subscript for FID and Clip score, I can only assume it's standard deviation, but FID measures the distance between two distributions so it's not supposed to have a standard deviation in it's natural form, but the paper didn't seems to explain that, additionally if the subscript serves as standard deviation, why the PSNR and SSIM do not have the standard deviation info.
4. Typos that affect the readability a lot. in line 110, it mentioned $w_t$. Since it has no meaning, I had to assume it's a typo and suppose it should be $x_t$ then continue reading, however in algorithm 1, in the Output line, $w_t$ appears again, now I'm not so certain if it's a typo or not.

**Questions:**

Most questions have been asked in the weakness section, here are some general questions for the author:
1. In Tables 1 and 2, ROBIN method achieves better verification time compared to Tree-ring. However, I'm wondering about the watermarking embedding time since adversarial optimization has been used. Does the optimization create a big bottleneck?
2. The watermarking capacity is not mentioned in the empirical section, in line 135, the author mentions that "we set the watermark as multiple concentric rings" but I didn't find how many concentric rings are exactly used.

-------------------------------------Post rebuttal Edition---------------------------------------

I appreciate the author's clarification. Most of my concerns have been addressed. With the potential readability enhancement in mind, I have changed my rating accordingly.

**Limitations:**

Limitations are adequately addressed. I think it's good research but the presentation may need heavy reformating to meet the standards of NeurIPS.

---

> ### Author Rebuttal · Authors · 2024-08-07
>
> > Q1. The optimization of $w_p$ and $w_i$ in lines 13 and 14 seems not to be consistent with Algorithm 1. The definition of $\epsilon_\theta(x_t^*,t,\psi(p),w_p)$ should be moved to the main text.
>
> (1) In the original manuscript, Lines 5 and 6 in Algorithm 1 indeed correspond to the minimization of the loss functions described in Equations 13 and 14. We recognize that the inclusion of additional inputs for the loss calculation in Algorithm 1 may have caused some confusion. Therefore, we will revise the expressions in Algorithm 1 to enhance clarity in the revised version.
>
> (2) We acknowledge the reviewer's suggestion and will move the definition of $\epsilon_\theta(x_t^*,t,\psi(p),w_p)$ from Appendix A.2 to the Methodology section for improved understanding.
>
>
>
>
> > Q2. Confusion for the caption and content of Figure 2. An illustration of the overlapped curve in Figure 2a.
>
> (1) The caption of Figure 2 in the original manuscript provides a simplified overview of the commonly used classifier-free guidance method [a], which has become a fundamental part of large-scale text-to-image diffusion models. A detailed explanation of this method has been presented in Appendix A.2 of the original manuscript.
> For better understanding, we include the definition from the original paper as follows. *For classifier-free guidance, the label $y$ in a class-conditional diffusion model $\theta(x_t|y)$ is replaced with a null label $\emptyset$ with a fixed probability during training. During sampling, the output of the model is extrapolated further in the direction of $\theta(x_t|y)$ (Condition) and away from $\theta(x_t|\emptyset)$ (Uncondition)as follows:* $Full=Uncondition + s\cdot (Condition - Uncondition)$. The second term of the addition is Guidance.
>
>
>
> Figure 2 (in the original manuscript) illustrates how the above components (Full, Uncondition, Condition and Guidance) change during the generation process (Figure 2a) and the impact of frequency domain perturbations on these components (Figure 2b). To enhance clarity, we will incorporate the relevant reference [a] within the Figure 2 caption and include the formula above in the revised version for better understanding.
>
> (2) We appreciate the reviewer's suggestion regarding Figure 2a. According to the above formula, the mean values of Uncondition and Condition are very close, resulting in their almost overlapping curves in Figure 2. We will incorporate a detailed explanation of the overlapped curves in the caption of the figure in the revised version.
>
> [a] GLIDE: Towards Photorealistic Image Generation and Editing with Text-Guided Diffusion Models. ICML, 2022.
>
> > Q3. Explanation about the subscript in Table 3 and the results for other metrics.
>
>
>
> (1) The subscripts in the original manuscript (Table 3) indicate the standard deviation of five independent experimental runs, each initialized with a different random seed. FID measures the difference between the five independently generated image sets and the ground-truth real image sets, so it has a standard deviation.
>
> (2) Due to space constraints, we have omitted the standard deviations for other metrics such as PSNR and SSIM. These values are provided in the table below.
> We will clarify the meaning of subscripts and include the standard deviations for all metrics in the revised version.
>
> Table 1. Image quality of five repeated experiments. Subscripts indicate standard deviation.
> | Model | Method | PSNR | SSIM | MSSIM |
> | :---- | :---- | :---- | :---- | :---- |
> | Stable Diffusion | Tree-Ring | ${15.37_{.07}}$ | $0.568_{.003}$ | $0.626_{.005}$ |
> |  | ROBIN | $24.03_{.04}$ | $0.768_{.000}$ | $0.881_{.001}$ |
> | Imagenet Diffusion | Tree-Ring | $15.68_{.03}$ | $0.663_{.002}$ | $0.607_{.001}$ |
> |  | ROBIN | $24.98_{.02}$ | $0.875_{.000}$ | $0.872_{.000}$ |
>
> > Q4. Some typographical errors.
>
>
>
> We thank the reviewer for identifying the typos. The variable $w_t$ in line 110 should be $x_t$. And $w_t$ in the output line and initializaion of Algorithm 1 should be $w_p$. These errors will be corrected in the revised manuscript. We will also conduct a thorough proofreading of the original manuscript to avoid any other typographical errors.
>
> > Q5. Does the optimization create a big bottleneck?
>
>
> (1) We do not believe watermark optimization to be a big bottleneck.
> First, the watermark optimization only needs to be done once. The optimized prompt embedding can then be applied universally to all images and text prompts without requiring further optimization. Second, the optimization process can be carried out offline, and the online watermark embedding incurs negligible overhead during image generation.
>
> (2) The table below presents the time costs per image for watermark optimization, watermarked image generation, and watermark verification using Stable Diffusion. The watermark optimization of ROBIN only incurs an average time overhead of 0.112 seconds per image. The time overhead for watermark embedding during image generation is 0.068 seconds, which is negligible compared to the average image generation time of 2.614s using Stable Diffusion. The verification of ROBIN watermarks is 80\% faster than Tree-Ring watermarks.
>
> Table 2. Time cost (s) of the watermark operation on each image.
> | Method | Optimization (Offline) | Generation (Online) | Validation |
> | :---- | :---- | :---- | :---- |
> | Stable Diffusion | - | 2.614 | - |
> | Tree-Ring | 0.000 | 2.617 | 2.599|
> |ROBIN | 0.112 | 2.682 | 0.531|
>
>
> > Q6. The watermarking capacity is not mentioned in the empirical section.
>
> As mentioned in Line 187 in the original manuscript, the watermark capacity is 70\% of the image frequency domain, corresponding to 30 and 120 concentric rings for Stable Diffusion and ImageNet diffusion model, respectively.

---

### Official Review · Reviewer_kSaH · 2024-07-16

**Soundness:** 3
**Presentation:** 2
**Contribution:** 3
**Rating:** 7
**Confidence:** 4

**Summary:**

This paper addresses watermarking in text-to-image diffusion models. Its main contributions are: 1) Embedding the watermark in the later stages of the diffusion process; 2) Introducing a text prompt guidance signal. These components collectively achieve better watermark robustness and image quality.

**Strengths:**

* Addresses a significant issue.
* Achieves a better trade-off between robustness and image quality compared to existing baselines.

**Weaknesses:**

* The robustness improvement over Tree-Ring is minor, only about 1% (see Table 1).
* Stable Signature is not compared in Table 2.
* Applicable only to diffusion models with DDIM sampler.
* Unclear how the proposed method resists frequency domain attacks.
* The rationale behind maximizing the watermark signal value to increase robustness is not explained.
* The initialization and optimization process for the text prompt w_p is unclear, especially when text prompt optimization is a discrete optimization problem.
* Why the original prompt used for image generation is unknown during verification?
* No noticeable improvement in the quality of generated images in terms of Clip score.

**Questions:**

See above.

**Limitations:**

See above.

---

> ### Author Rebuttal · Authors · 2024-08-07
>
> > Q1. The robustness improvement over Tree-Ring is minor.
>
> (1) Compared to Tree-Ring watermarks, ROBIN improves the robustness from 0.975 to 0.983 on Stable Diffusion, effectively reducing the error rate by 32\% (from 0.025 to 0.017). Achieving further improvements on an already high AUC of 0.975 is inherently challenging.
>
> (2) Additionally, our robustness evaluations were conducted under severe attacks that significantly compromised image quality, making watermark verification much more difficult. Thus, we believe our improvement is significant.
>
> (3) Moreover, ROBIN offers substantial advantages in verification speed (80\% improvement in verification time) and image quality (35\% improvement in SSIM), compared to Tree-Ring watermarks. These enhancements underscore the substantial benefits of our approach.
>
> > Q2. Stable Signature is not compared in Table 2.
>
> As mentioned in Line 198 of the original manuscript, Stable Signatures are specifically designed for latent diffusion models and are therefore incompatible with pixel-level ImageNet diffusion models (Table 2). This limitation is also mentioned in the Introduction section of the original paper of Stable Signatures [a].
>
> [a] The stable signature: Rooting watermarks in latent diffusion models. ICCV, 2023.
>
> > Q3. Applicable only to diffusion models with DDIM sampler.
>
> The applicability of ROBIN is not limited to DDIM samplers. Our watermark verification requires a reversible generation process, making it compatible with any reversible samplers such as DPM-Solver [a], DPM-Solver++ [b], PNDM [c], and AMED-Solver [d]. Our experiments employed both DPM-Solver and DDIM, and we anticipate ROBIN's adaptability to future reversible generation algorithms.
>
> [a] Dpm-solver: A fast ode solver for diffusion probabilistic model sampling in around 10 steps. NeurIPS, 2022.
>
> [b] Dpm-solver++: Fast solver for guided sampling of diffusion probabilistic models. arXiv:2211.01095 (2022).
>
> [c] Pseudo Numerical Methods for Diffusion Models on Manifolds." ICLR, 2022.
>
> [d] Fast ode-based sampling for diffusion models in around 5 steps. CVPR, 2024.
>
> > Q4. Unclear how the proposed method resists frequency domain attacks.
>
> Following your suggestions, to further assess robustness, we evaluated ROBIN under various low-pass filtering frequency attacks, which interfere with the frequency domain of the image without destroying the main content. The accompanying table presents AUC values on Stable Diffusion under different attack methods, demonstrating the superior robustness of our approach to frequency domain attacks.
>
>
> Table 1. Performance evaluation under frequency-domain attacks.
> | Method | Ideal Low-pass | Butterworth Low-pass | Gaussian Low-pass |
> | :---- | :---- | :---- | :---- |
> | StableSig | 0.879 | 0.932 | 0.933 |
> | Tree-Ring | 0.975 | 0.999 | 0.996 |
> | ROBIN | **0.987** | **0.999** | **0.999** |
>
> > Q5. The rationale behind maximizing the watermark signal value to increase robustness.
>
> (1) Our approach adheres to the fundamental principle that stronger signals exhibit greater resilience to noise and interference. Existing attacks typically add small noise signals to the original image to disrupt the watermark without damaging the image. A stronger watermark signal has more "power" to overpower the noise signal, as demonstrated in [a].
>
> (2) In the context of frequency domain watermarking that we use, the strength of the watermark signal is positively correlated with its numerical value. By maximizing the value of the watermark, we increase the watermark strength, thereby improving robustness.
>
>
> [a] Digital watermarking and steganography. Morgan kaufmann, 2007.
>
>
> > Q6. The initialization and optimization process for the text prompt $w_p$ is unclear, especially when text prompt optimization is a discrete optimization problem.
>
>
> (1) As stated in line 438 of the original manuscript, we initialized $w_p$ as empty strings to minimize its interference with the generation process.
>
>
> (2) In our approach, we optimize a continuous prompt embedding rather than the discrete text prompt, which allows it to be optimized via gradient descent.
>
> Model owners can introduce $w_p$ into the generation process as another guidance term independent of the original text for watermarking.
> We will clarify it in the revised version and replace the words ''prompt $w_p$'' with ''prompt embedding $w_p$'' for clarity.
>
> > Q7. Why the original prompt used for image generation is unknown during verification?
>
> We target a more general watermark verification scenario where users might generate images using diffusion models and publish them online. Our goal is to determine whether these published images are authentic, even when users do not provide their diffusion prompts (likely they won't).
>
>
> > Q8. No noticeable improvement in the quality of generated images in terms of Clip score.
>
> (1) In our paper, the image quality is assessed by PSNR, SSIM, and MSSIM metrics rather than CLIP score. And we achieve an improvement of 35\% in SSIM over Tree-Ring watermarks.
>
> (2) CLIP score evaluates the alignment between the generated image and its corresponding text prompt and is constrained by the base model's generative capacity. We achieved a relative 8.8\% improvement in CLIP score (from 0.364 to 0.396) on Stable Diffusion (SD) compared to the optimal Tree-Ring watermarks, nearing the upper bound of the base model (SD) with a CLIP score of 0.403.

---

> > ### Comment · Reviewer_kSaH · 2024-08-14
> > **Thanks for the response!**
> >
> > I think the response addresses most of my concerns and I decided to raise my score. Thanks.

---

### Official Review · Reviewer_Axd1 · 2024-07-17

**Soundness:** 3
**Presentation:** 3
**Contribution:** 3
**Rating:** 6
**Confidence:** 4

**Summary:**

This paper aims to balance robustness and concealment for image watermarking generated by diffusion models. The authors propose a novel method that actively hides stronger watermarks while ensuring their imperceptibility. They introduce a two-step process: first, embedding a robust watermark in intermediate diffusion time-steps, then using an adversarial optimization algorithm to generate a "hiding prompt" that guides the model to conceal the watermark in the final image. This approach aims to maximize watermark strength while minimizing visual artifacts. The proposed method has been evaluated on both latent and image diffusion models.

**Strengths:**

Originality: The work builds upon the existing tree-ring watermarks for diffusion models [39]. The idea of simultaneously optimizing the prompt for stealthiness and the watermark for robustness is particularly innovative in an adversarial manner is interesting. This method also does not require training diffusion model parameters, unlike previous techniques, and it is shown to be more robust than the tree-ring-based baseline. Furthermore, the authors claim that their approach preserves the semantics of the original stable diffusion model, unlike the tree-ring-based method.

Quality: The authors clearly explain their methodology, particularly the introduction of a watermark hiding process and the use of an adversarial optimization algorithm. The study includes a thorough comparative analysis with five baselines, including those based on diffusion models, demonstrating the robustness of the proposed method against various image transformations.

Clarity: The paper is generally well-organized and clearly written.

Significance: By addressing the crucial balance between robustness and concealment, the paper tackles a major challenge in digital watermarking of diffusion models. The method's ability to embed stronger watermarks while maintaining imperceptibility could have far-reaching implications for content protection. Moreover, the approach's compatibility with existing diffusion models without requiring retraining enhances its practical significance and potential for widespread adoption.


[39] Tree-ring watermarks: Fingerprints for diffusion images that are invisible and robust. In Advances in Neural Information Processing Systems, 2023.

**Weaknesses:**

1 - Motivation for Preserving Semantics (Line 37): The author states that the tree-ring watermark baseline [39] leads to semantic changes that their method does not. However, the motivation behind this is unclear. Although the tree-ring watermark approach results in slight semantic alterations compared to the original stable diffusion model, the changes remain faithful to the original text prompt with a negligible drop in the FID score. This implies that the user experience is not significantly affected. Therefore, it would be beneficial to clarify the rationale behind the emphasis on preserving the semantics of the original image and how it contributes to the overall goal of the watermarking method.

2 - Attribution for frequency domain embedding: On line 133, the statement "To achieve robustness, we embed the watermark in the frequency domain of the image" should properly credit the tree-ring watermark baseline [39].

3 - Watermark validation threshold: In section 3.4, the authors do not explain how they selected the L1 distance threshold for watermark verification. The authors should explicitly describe the methodology for determining this threshold, including any empirical studies.

4 - Fairness of comparison metrics: Table 3's comparison using PSNR, SSIM, and MSSIM may not be entirely fair when comparing to the tree-ring watermark baseline, which doesn't aim to preserve semantics relative to the original diffusion model, although still faithful to the text prompt. For example, given a prompt "a white dog", the tree-ring method will generate a white dog, but it may differ from the output of the original diffusion model (just like changing random seed). Instead, using the FID score, which assesses the similarity of generated images to real images, would be a more appropriate metric. The proposed method shows relatively poor performance in terms of FID, and this should be addressed to provide a more balanced and accurate evaluation of the method's effectiveness.

[39] - Tree-ring watermarks: Fingerprints for diffusion images that are invisible and robust. NeurIPS, 2023.

**Questions:**

Please refer to the weakness section.

**Limitations:**

yes

---

> ### Author Rebuttal · Authors · 2024-08-07
>
> > Q1. Clarify the motivation for preserving semantic, given that the sematic alterations caused by Tree-Ring remain faithful to the original text prompt, as evidenced by a negligible drop in FID.
>
> (1) FID cannot evaluate the semantic faithfulness of the generated images to the orignal textual prompt. FID only measures the distributional distance between two unordered sets of images at the pixel level, regardless of the semantic content. So, a minor decrease in FID does not ensure that the generated images remain semantically consistent with the text.
>
> (2) Semantic modifications induced by Tree-Ring watermarks are random and may result in images that deviate from the original textual prompts. This is evidenced by the decline in the CLIP score by 10\% (from 0.403 to 0.364) on Stable Diffusion, which quantifies the correspondence between generated images and the given text. Figure 3 in the original manuscript presents examples of such generation failures.
>
> (3) Therefore, we aim to exactly preserve the original semantics to achieve a better lower bound for faithfulness. We also improve the CLIP score by 8.8\% compared to Tree-Ring. By generating a watermarked image semantically aligned with the original one, we minimize its impact on user experience (image-text alignment).
>
> (4) In addition, we potentially support the scenario when users may expect watermarked and original Stable Diffusion to produce the same outputs to verify that they are using the correct version, which necessitates semantic preservation.
>
>
> > Q2. Attribution for frequency domain embedding.
>
> In the original manuscript, we have credited Tree-Ring watermarks for the frequency domain embedding in lines 134 and 135. But for clarity, we will move it to line 133 in the revised version.
>
>
> > Q3. How to select the L1 distance threshold for watermark verification.
>
> (1) In practical application scenarios, a fixed threshold is required for per-image watermark detection. We empirically set this threshold as the median L1 distance between 250 watermarked and 250 non-watermarked images. Using the calculated threshold (31.45), we achieve 100\% detection accuracy on a separate 1000-image dataset.
>
>
> (2) For research comparisions, we aim for a thorough evaluation under various thresholds to ensure a larger effective threshold interval. Therefore, we follow Tree-Ring [a] and use the Area Under the Curve (AUC) metric. AUC represents the area under the Receiver Operating Characteristic (ROC) curve, which plots the fraction of true positive results against the fraction of false positive results at various threshold settings. A higher AUC indicates a broader range of effective thresholds for real-world applications, reflecting a higher tolerance for errors.
>
> [a] Tree-ring watermarks: Fingerprints for diffusion images that are invisible and robust. NeurIPS, 2023.
>
>
> > Q4. Metrics like SSIM may not be entirely fair when comparing with Tree-Ring which doesn't aim to preserve semantic but keep faithfulness. Instead, FID would be a more appropriate metric.
>
>
>
> (1) We acknowledge that in generative watermarking, it is not necessary for the watermarked image to exactly match the original image. Ideally, we may find another watermarked image that aligns with the text prompt, even if it differs from the original image. However, achieving this is challenging. Tree-Ring watermarks, as noted in response to Q1, are more akin to random semantic modifications and do not guarantee the same level of text alignment as the original generation (10\% reduction in CLIP score).
>
> (2) Instead, we make the simplest assumption that \textbf{preserving the original semantics provides a better lower bound for faithfulness}. Thus, we used PSNR, SSIM, and MSSIM to evaluate the similarity between the images before and after adding watermarks. Our improvement lies in achieving stronger robustness than Tree-Ring watermarks while producing outputs nearly identical to the original image.
>
> (3) Additionally, our method potentially supports adding watermarks to a given image as we can maintain almost identical outputs. In contrast, Tree-Ring watermarks are limited to the generation phase.
>
> (4) As we mentioned in the response for Q1, FID cannot evaluate the semantic faithfulness of the generated images to the original textual description but CLIP score does.
>
> In addition, FID is only calculated on pixel level and is a less reliable metric for semantic integrity, which is crucial in generative watermarking.
> The table below illustrates the impact of various pixel-level image transformations on FID, including random erasing with ratio 7\%, center cropping of 80\%, JPEG compression with quality 45 and random rotation of 30 degree. The processed image samples are provided in Figure 1 in the rebuttal pdf.
>
> The results show that JPEG compression, causing negligible visual changes for huamn eyes, significantly affects FID. In contrast, randomly erasing 10\% of the image, which substantially alters image semantics, has limited influence on FID.
>
> Table 1. FID under different image transformations.
> |  | Original | Erase | Crop | JPEG | Rotate  |
> | :---- | :---- | :---- | :---- | :---- | :---- |
> | FID | 25.53 | 25.88 | 26.96 | 27.42 | 29.90 |

---

> > ### Comment · Reviewer_Axd1 · 2024-08-13
> >
> > The authors have addressed most of my concerns, so I have raised the score. I hope the authors will open-source their code for reproducibility.

---

> > > ### Author Response · Authors · 2024-08-14
> > > **Thanks for your comment!**
> > >
> > > Dear Reviewer Axd1, thanks for recognizing our responses. We are happy that our response has addressed your concerns. We will publish the code base for all the experiments with the camera-ready version. Thank you again for your thoughtful review and support.

---

### Author Rebuttal · Authors · 2024-08-07

We sincerely thank the anonymous reviewers for their valuable and constructive comments and suggestions. And some figures are contained in the attached pdf.

---

### Decision · Program_Chairs · 2024-09-25

**Decision:**

Accept (poster)

**Comment:**

The paper initially received largely positive reviews: 2 BA, 1 WA, and 1 BR. The major concerns raised were:

1) unclear motivation and descriptions (Axd1, kSaH)
2) fairness to compare with tree-ring method (Axd1)
3) incremental improvement over tree-ring method (kSaH)
4) no noticeable improvement of CLIP score (kSaH)
5) limited to DDIM samplers? (kSaH, 4gSJ)
6) Readability problems (duRp, 4gSj)
7) include stronger attacks (4gSJ)

The authors wrote a response, and afterwards all reviewers were satisfied and increased their ratings. In the end, all reviewers were positive. The paper should be updated according to the reviews and discussion.